# Transapical Approach to Septal Myectomy for Hypertrophic Cardiomyopathy

**DOI:** 10.3390/life14010125

**Published:** 2024-01-15

**Authors:** Alexander Afanasyev, Alexander Bogachev-Prokophiev, Sergei Zheleznev, Mikhail Ovcharov, Anton Zalesov, Ravil Sharifulin, Igor’ Demin, Bashir Tsaroev, Vladimir Nazarov, Alexander Chernyavskiy

**Affiliations:** Federal State Budgetary Institution National Medical Research Center Named after Academician E.N. Meshalkin of the Ministry of Health of the Russian Federation, Rechkunovskaya Str., 15, 630055 Novosibirsk, Russia

**Keywords:** hypertrophic cardiomyopathy, left ventricular outflow obstruction, apical myectomy, heart failure, cardiac transplantation

## Abstract

A 63-year-old symptomatic female with apical hypertrophic cardiomyopathy and diastolic disfunction was admitted to the hospital. What is the best way to manage this patient? This study is a literature review that was performed to answer this question. The following PubMed search strategy was used: ‘Hypertrophic obstructive cardiomyopathy’ [All Fields] OR ‘apical myectomy’ [All Fields], NOT ‘animal [mh]’ NOT ‘human [mh]’ NOT ‘comment [All Fields]’ OR ‘editorial [All Fields]’ OR ‘meta-analysis [All Fields]’ OR ‘practice-guideline [All Fields]’ OR ‘review [All Fields]’ OR ‘pediatrics [mh]’. The natural history of the disease has a benign prognosis; however, a watchful strategy was associated with the risk of adverse cardiovacular events. Contrastingly, transapical myectomy was associated with low surgical risk and acceptable outcomes. In our case, the patient underwent transapical myectomy with an unconventional post-operative period. Control echocardiography showed marked left ventricular (LV) cavity enlargement: LV end-diastolic volume, 74 mL; LV ejection fraction, 65%; and LV stroke volume index increased to 27 mL/m^2^. The patient was discharged 7 days after myectomy. At 6 months post-operation, the patient was NYHA Class I, with a 6 min walk test score of 420 m. Therefore, transapical myectomy may be considered as a feasible procedure in patients with apical hypertrophic cardiomyopathy and progressive heart failure.

## 1. Introduction

Hypertrophic cardiomyopathy (HCM) is the most common hereditary heart disease, affecting 1:200–1:500 of the general population [1,2]. The pathophysiology of HCM includes progressive symptoms of left ventricular outflow tract obstruction (LVOTO), progressive diastolic dysfunction, systolic-dysfunction-induced heart failure, atrial fibrillation with a risk of stroke, sustained ventricular tachycardia, and sudden cardiac death (SCD) events [3]. There are several variants of obstructive HCM, including sigmoid septal morphology, mid-cavity obstructive HCM, right ventricular obstruction, apical form, and end-stage HCM. Reportedly, the first three variants are routinely scheduled for a septal myectomy procedure with Class I recommendations as per the current guidelines [3], and patients with end-stage HCM may be considered for heart transplantation. However, patients with apical HCM have been reported to have a favorable and benign prognosis regarding cardiovascular mortality and SCD; therefore, they are not considered as surgery candidates [4,5]. Nevertheless, recent studies have shown that apical HCM is associated with increased mortality, particularly in the older population and women [6,7]. Furthermore, the “mixed” form, with predominantly apical hypertrophy and thickening of the non-apical left ventricle region, has a worse prognosis for cardiovascular death and adverse events than pure apical HCM [8].

## 2. Materials and Methods

### 2.1. Case Scenario

A 63-year-old female with marked limitations in daily activities, fatigue, and shortness of breath while walking a short distance (approximately 50 m) was admitted to the hospital and diagnosed as New York Heart Association (NYHA) class III-IV. An electrocardiogram showed high QRS voltage and negative T waves in the precordial leads. Transthoracic echocardiography revealed an apical HCM variant with a left ventricular end-diastolic volume (LVEDV) of 59 mL, increased left atrial volume index of 44 mL/m^2^, average E/e’ > 14, LV ejection fraction (LVEF) of 69% without LVOTO (maximum exercise provokable LVOT gradient 15 mmHg), a reduced stroke volume (SV) index of 22 mL/m^2^, Grade 1 (mild) mitral regurgitation findings, and no systolic anterior motion syndrome. There was no evidence of subaortic obstruction. The maximum septal thicknesses in the basal, midventricular, and apical parts were 15, 15, and 19 mm, respectively (Figure 1, Appendix A). Coronary angiography revealed a right-dominant circulation without pathological outlines. A cardiac MRI confirmed apical HCM with maximum septal thicknesses in the basal, midventricular, and apical parts of 9, 14, and 20 mm, respectively; maximum LV lateral wall thicknesses of 6, 6, and 21 mm, respectively; maximum LV posterior wall thicknesses of 5, 6, and 18 mm, respectively; and maximum LV anterior wall thicknesses of 6, 6, and 18 mm, respectively (Figure 2, Appendix A).

There were no clinical evidence or CMR contrast imaging data suggesting amyloidosis, Fabry disease, or any other lysosomal storage disease in this case.

Current medications, such as non-vasodilating beta-blockers, were ineffective. According to the current guidelines, cases wherein LVOTO and attributive target septal thickness are absent are not candidates for a Morrow procedure or transaortic extended septal myectomy. Such patients may be considered for pharmacological management using non-dihydropyridine calcium channel blockers. However, is there evidence that surgical strategies will have a clinical benefit in this case?

According to our literature review (Table 1 and Table 2) and based on data of severely symptomatic patients’ status (NYHA Class III), a high risk of progression of LV diastolic dysfunction potentially causes a heart transplant waitlist, additional risk factors of major events, and mortality (older age, female sex). In our case, after shared decision-making, including the risks and benefits of all treatment options, transapical septal myectomy was considered an alternative to escalation to medical therapy. Written informed consent was obtained from the patient for this exclusive surgery and for anonymized information to be published in this article. This case report study conformed to the tenets of the Declaration of Helsinki.

### 2.2. Surgical Procedure

We have experience of performing over 500 septal myectomies [15] and many concomitant procedures [16]; however, we do not have expertise in transapical myectomy. We perform up to 100 transaortic procedures annually; however, every mildly symptomatic patient with apical HCM is considered for pharmacological management owing to the presumed benign prognosis. Therefore, we consulted Dr. H. Scaff from the Mayo Clinic regarding the technical details of the surgical procedure and the potential pitfalls of transapical myectomy. We adopted the surgical technique described by Kotkar et al. [9] and performed a transapical approach for myectomy through median sternotomy, central aortic and two-stage venous cannulation, and instituted normothermic cardiopulmonary bypass. Myocardial protection was achieved using antegrade normothermic blood cardioplegia with a perfusion blood flow of 300 mL/min and a bolus of 5 mL, followed by a flow of 30 mL of Calafiore’s cardioplegia solution from the perfusion for 2 min.

A few gauze napkins were placed under the LV to ensure adequate apex exposure. An apical ventriculotomy was performed lateral to the left anterior descending artery. The LV apex was obliterated by the hypertrophied LV walls, and papillary muscles were displaced apically; however, they were not injured while the LV apex was opened. Myectomy was performed by sequentially removing the hypertrophied muscle mass and shaving the anterolateral and posterior LV walls using a No. 11 blade knife and Valve XS micro scissors (Aesculap, Melsungen, Germany) with Zenker retractors (Aesculap, Melsungen, Germany) (Figure 3, Appendix A). The primary focus of the procedure was to achieve LV cavity augmentation, shave hypertrophied muscles proximal to the enlarged chamber, and reduce the risk of residual obstruction [9]. The adequacy of resection was controlled through “ad oculus” and index finger palpation. To minimize the risk of systemic embolism, the LV cavity was irrigated and debris was removed using a vacuum aspirator. A direct-view control inspection was performed to ensure that the mitral valve apparatus was not damaged. The absence of a ventriculus septum defect was assessed by filling the right chamber. The ventriculotomy was closed using felt strips and two layers of running mattress sutures of number 3/0 Premilene. The cross-clamp and overall bypass times were 22 and 43 min, respectively. There were no residual mitral regurgitation, LVOTO, ventricular septal defects, or other procedure-related complications on intraoperative control transesophageal echocardiography.

## 3. Results

The post-operative period was unconventional. Post-operative transthoracic echocardiography showed marked LV cavity enlargement compared with the baseline data: LV EDV, 74 mL; LV EF, 65%; and LV SV index increased to 27 mL/m^2^ (Figure 4, Appendix A). The maximum LVOT gradient provoked by the exercise was 6 mmHg. A control cardiac MRI underlined the achievement of LV augmentation: LV EDV, 86.9 mL and LV EF, 64%. The maximum septal thicknesses in the basal, midventricular, and apical parts were 11, 10, and 7 mm, respectively; the maximum LV lateral wall thicknesses were 7, 7, and 3 mm, respectively; the maximum LV posterior wall thicknesses were 9, 8, and 9 mm, respectively; and the maximum LV anterior wall thicknesses were 8, 7, and 7 mm, respectively (Figure 5, Appendix A). The patient was discharged to the rehabilitation facility 7 days after surgery with a stable sinus rhythm. Furthermore, 6 months after the transapical myectomy, the patient was in NYHA Class I, with a 6 min walk test score of 420 m.

## 4. Discussion

Pathophysiological phenomena in patients with apical HCM include mid-ventricular obstruction with cavity obliteration and apical aneurysm formation. The clinical manifestations of apical HCM include dyspnea, exercise intolerance, and pulmonary edema due to diastolic dysfunction, which leads to increased filling pressures and left atrial enlargement.

Myocardial ischemia, which develops in the absence of epicardial atherosclerotic disease, is associated with small vessel disease with intramural coronary artery narrowing.

The worst complication of apical HCM is arrhythmogenic SCD caused by sustained ventricular tachycardia and/or ventricular fibrillation. The cardioverter-defibrillator may be an effective strategy for the prevention of SCD in patients with high-risk apical HCM.

Apical myectomy is a novel surgical approach, which includes the removal of hypertrophic muscle from the apical and mid-left ventricle to improve functional status and SV. Apical myectomy is attempted in patients who continue to experience unabated symptoms and severe lifestyle limitations such as diastolic heart failure, NYHA functional Class III–IV dyspnea despite optimized drug therapy, and in cases where apical myectomy may be the only alternative to heart transplantation.

Apical HCM is unassociated with cardiac death and has a benign prognosis regarding cardiovascular mortality. However, many patients experience critical cardiovascular complications, such as myocardial infarction and arrhythmia. Septal myectomy is disappearing in westernized countries, giving way to less invasive procedures such as alcohol septal ablation. Nevertheless, in high-volume HCM centers, septal myectomy is a gold standard procedure; meanwhile, the less invasive alcohol septal procedure is only an alternative option for high-risk patients that is consistent with current ESC and ACC/AHA guidelines. Furthermore, we do not recommend alcohol septal ablation for apical HCM.

We would emphasize that HCM with basal obstruction is quite different from apical HCM. In the first case, severe symptoms are related to LVOT obstruction. In pure apical HCM patients, the cause of clinical manifestations is the development of severe LV diastolic dysfunction. Transaortic myectomy and transapical myectomy are more than just two approaches to the septum, they are different surgical procedures for different goals. The first one relieves LVOTO; the second one is the LV enlargement procedure.

The first-in-class inhibitor of cardiac myosin, Mavacamten, reduces contractility and improves cardiac function. The EXPLORER-HCM trial reported that Mavacamten improved NYHA functional class, health status, functional capacity, and LVOT obstruction in a representative HCM population. Nevertheless, there is no clear recommendation for using this novel drug therapy including cardiac myosin inhibition, perhexiline, and gene therapy for patients with apical HCM. Meanwhile, we believe that future trials should address this issue.

There is no consensus on managing patients with apical HCM. Several studies [10,11,12] have proven that apical myectomy is safe and effective for treating symptomatic patients. Many currently published papers are observational studies without comparison groups and that lack long-term outcomes.

### 4.1. Surgery

Apical myectomy is an option for patients with persisting unabated symptoms and severe lifestyle limitations (diastolic heart failure, NYHA functional Class III–IV dyspnea) following fully optimized drug therapy, making surgery the only likely alternative to heart transplantation.

According to Kotkar et al.’s [9] retrospective study in 2000, the purpose of apical myectomy was to increase the left ventricular end-diastolic and systolic dimensions, which would increase stroke volume (SV) to the normal range. All patients with obstruction experienced gradient relief, and none developed ventricular septal defects. There were no complications from the left ventriculotomy, such as left ventricular apical aneurysm or ventricular arrhythmias.

Nguyen et al. [11] performed a retrospective study with a 10-year follow-up. The survival rates at 1, 5, and 10 years were 96%, 87%, and 74%, respectively. Similarly, 76% of patients reported improved symptoms, and three (3%) subsequently underwent cardiac transplantation for recurrent heart failure. Their survival was superior to that of patients with HCM listed for heart transplantation.

In their retrospective study including six patients, Forteza et al. [12] reported that two patients underwent heart transplantation. Echocardiographic evaluation revealed a significantly increased ventricular volume in all patients. The primary conclusion was that apical myectomy is a safe and effective treatment and could offer an alternative to heart transplantation in patients with advanced heart failure. Similarly, some studies [10,13] gave excellent results in the early and mid-term periods regarding gradient relief, increased LVEDV, and improved diastolic function and SV. Patients with apical HCM with symptoms despite optimal medical treatment and apical myectomy may have improved hemodynamics and ameliorating symptoms [13].

In 2002, Eriksson et al. [5] evaluated the prognosis of 105 patients, with a follow-up of 13.6 ± 8.3 months. The probability of survival without morbid events at 15 years was 74%. The most frequent morbidities in this study were atrial fibrillation and myocardial infarction. Atrial fibrillation occurred in 12% of patients. Apical myocardial infarction in the presence of normal coronary arteries was found in approximately 10% of patients. Despite this significant morbidity, most patients showed no deteriorations in their functional class, and approximately half remained totally asymptomatic during follow-up.

Webb et al. [4] studied 26 patients with a follow-up of 7.3 ± 6.2 years, 10 of which were asymptomatic. In the remaining 16 patients, the major symptoms included atypical chest pain (n = 10), angina (n = 6), palpitations (n = 8), exertional dyspnea (n = 5), presyncope (n = 4), and fatigue (n = 4). Some patients exhibited more than one symptom. During follow-up, one patient without a history of angina had myocardial infarction with aneurysm formation. Two other patients developed worsening angina, one of whom had coronary disease and a non-Q wave infarction.

In addition, Towe et al. [14] reported on approximately 71 patients with a mean follow-up of 5.5 years. Apical aneurysms that were structurally absent upon initial diagnosis were identified in two patients (3%) at follow-up. Twenty-nine patients (43%) underwent implantable cardioverter defibrillator or pacemaker implantation. Upon review of the data for adverse outcomes, as previously defined, 24% of patients with apical HCM (16 of 68) were classified as having NYHA Class III or Class IV heart failure.

In 2013, Klarich et al. [6] evaluated 193 patients with a mean follow-up period of 78 months (range, 1–350). All-cause deaths occurred in 55 patients (29%, 33 women). During follow-up, more women had heart failure (*p* = 0.001), atrial fibrillation (*p* = 0.009), or died (*p* < 0.001) than men. Survival was worse than expected (*p* = 0.001); the observed versus expected 20-year survival rates were 47% and 60%, respectively. SCD, resuscitated cardiac arrest, and/or defibrillator discharge were observed in 11 (6%) patients during follow-up.

Moon et al. [7] evaluated 454 patients with a mean follow-up period of 43 ± 20 months. Major adverse cardiovascular events occurred in 110 patients (25%) of the enrolled population. Thirty-nine patients (9%) died during the monitoring period. Six deaths (1%) were suspected to be caused by SCD. Stroke- and heart-failure-induced deaths occurred in 7 (2%) and 11 (2%) patients, respectively. A total of 88 patients (19%) were hospitalized for heart failure management, and 26 (6%) experienced ischemic stroke during monitoring. Nonfatal myocardial infarction occurred in two patients without significant coronary artery disease. Patients with apical HCM and poor clinical outcomes had more advanced diastolic dysfunction, reduced myocardial contraction/relaxation properties, and increased LV filling pressure at presentation.

### 4.2. Clinical Bottom Line

Apical HCM is a complex subset of HCM with highly heterogeneous clinical symptoms and pathological profiles, and predominantly non-obstructive physiology. Patients with apical HCM had good health and an active lifestyle. In contrast, those with apical HCM had dyspnea and/or angina (refractory to pharmacotherapy) and SCD. The prognosis appeared relatively favorable in most patients with apical HCM, and there were adverse events, such as myocardial infarction, atrial fibrillation, and advanced heart failure. According to recent data, surgical treatment is unassociated with high operative mortality or complications. However, owing to the limited number of controlled studies and their retrospective designs, the results should be confirmed in prospective studies with a large patient population.

Thus, physicians should consider referring patients to centers with expertise in HCM management to choose the best treatment technique.

## 5. Study Limitations

This was a case report with a literature review. There are no randomized trials or prospective studies directly comparing transapical myectomy and a watchful strategy for the management of patients with apical HCM. The available literature is limited by the case report series, single-center, and retrospective nature of the studies. Therefore, there is not enough evidence to perform a systematic review and meta-analysis to obtain a strong recommendation in this field. Further studies will be needed to confirm our assumptions relating to transapical myectomy as the best strategy for patients with apical HCM and severe diastolic disfunction.

Heart team discussed the treatment of this symptomatic patient. The final decision for the treatment in this case was supported by the suggestions offered by the American Hypertrophic Cardiomyopathy guidelines as it is categorized as 2B. However, we get the approval of the hospital ethics committee for this type of operation supported by the heart failure progression (not related to other causes).

## Figures and Tables

**Figure 1 life-14-00125-f001:**
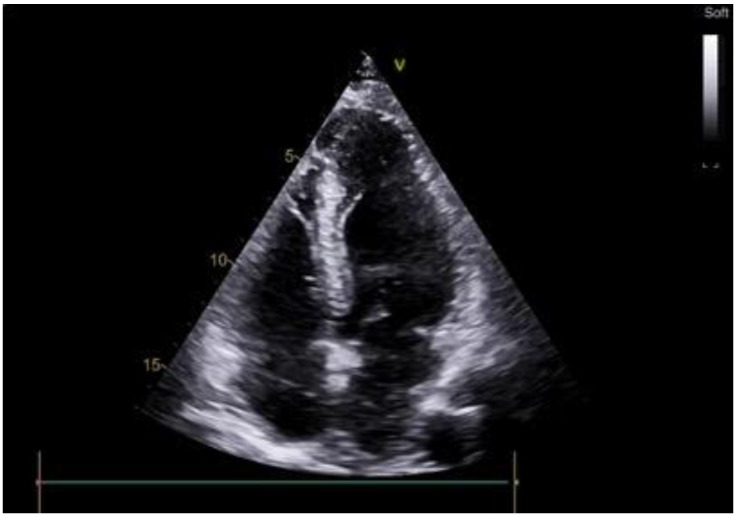
Preoperative echocardiography of patient with apical hypertrophic cardiomyopathy.

**Figure 2 life-14-00125-f002:**
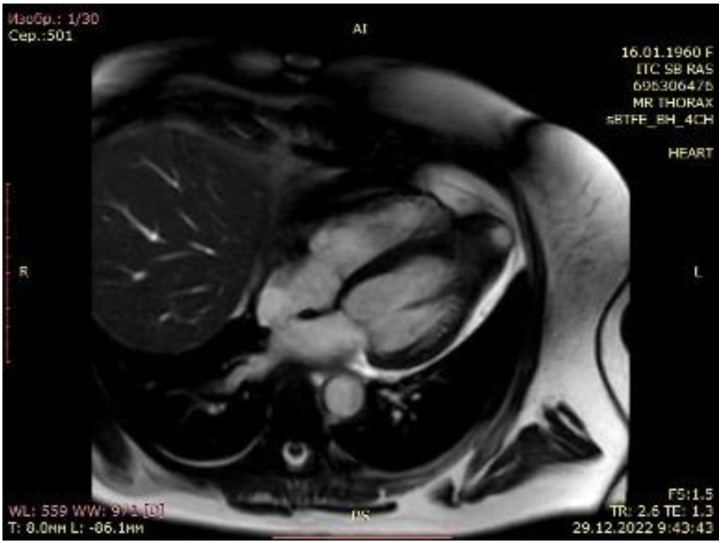
Cardiac magnetic resonance imaging of patient with apical hypertrophic cardiomyopathy.

**Figure 3 life-14-00125-f003:**
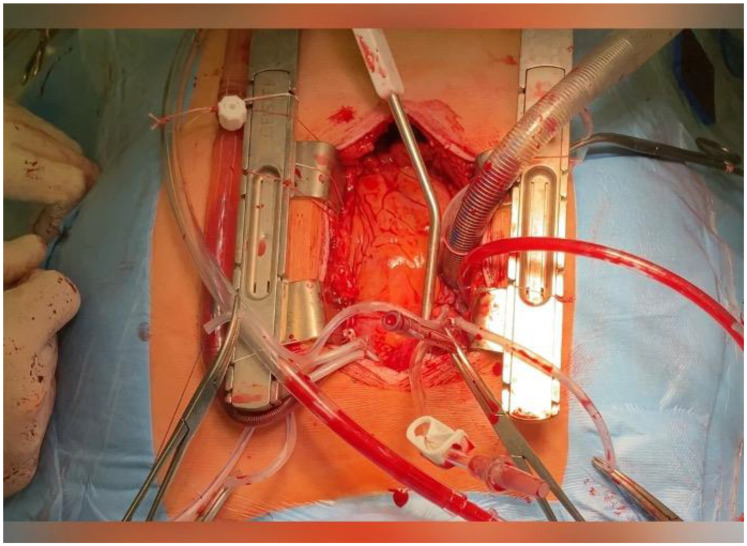
Intraoperative view of transapical myectomy.

**Figure 4 life-14-00125-f004:**
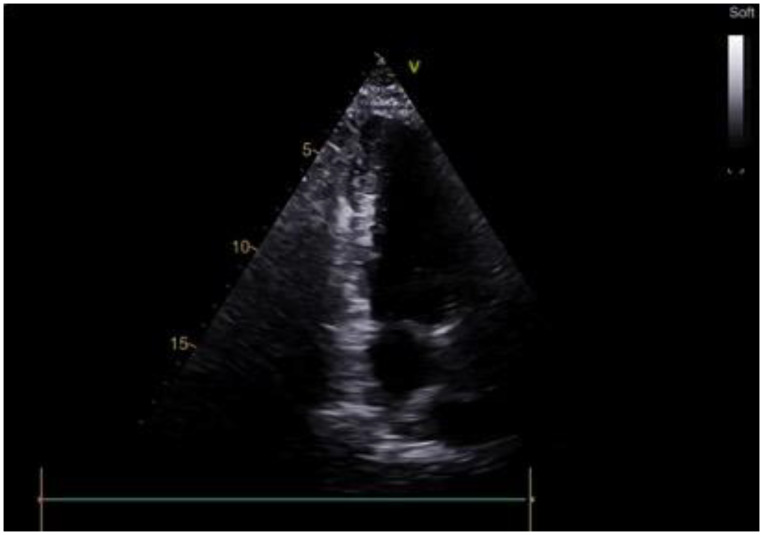
Echocardiography of patient with apical hypertrophic cardiomyopathy after transapical myectomy.

**Figure 5 life-14-00125-f005:**
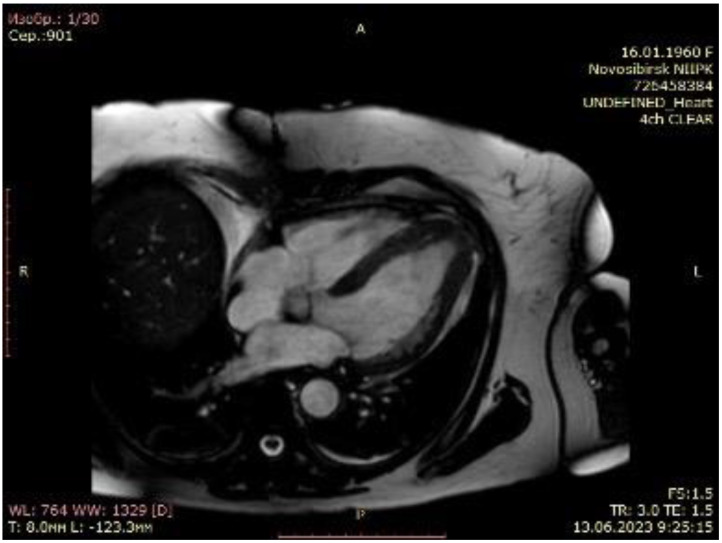
Cardiac magnetic resonance imaging of patient with apical hypertrophic cardiomyopathy after transapical myectomy.

**Table 1 life-14-00125-t001:** Myectomy for apical hypertrophic cardiomyopathy, review *.

Author, Date, Journal, Country, and Study Type (Level of Evidence)	Patient Group	Outcomes	Key Results	Comments
Kotkar et al. [9], Ann Cardiothorac Surg Retrospective study (Level 2b)	55 patients with an obstructive midventricular level and 60 patients with midventricular and apical variants of HCM. There were no patients with ventricular aneurysm and ventricular tachycardia.	Clinical outcomes	All patients had gradient relief. None of the patients developed an apical aneurysm or ventricular septal defect.	Retrospective nature of the study
Shimizu et al. [10], Asian Cardiovasc Thorac Ann. Cohort study (Level 2b)	59 patients Age: 40 (39.3–60.7) NYHA class: 16 (27.4%) III/IV Apical Aneurysm n (%): 13 (22.0)	30-day mortality; 5-year survival	Two perioperative deaths, one late death caused by acute myocardial infarction. 95% The intraventricular gradient had significantly decreased at discharge, and no reoperation for recurrent obstruction was conducted.	Single-center case series
Nguyen et al. [11], J Thorac Cardiovasc Surg Retrospective study (Level 2b)	113 symptomatic patients: Male: 49 (43%) Age: 50.8 (39.3–60.7) NYHA class: 108 (96%) III/IV Ventricular tachycardia n (%): 20 (18) Apical Aneurysm n (%): 25 (22)	30-day mortality 1-year survival 5-year survival 10-year survival Clinical symptoms	4 (4%) deaths 96% 87% 74% 76% of patients reported improvement in symptoms, and 3 patients (3%) subsequently underwent cardiac transplantation for recurrent heart failure. Survival appeared superior in patients with hypertrophic cardiomyopathy listed for heart transplant.	Retrospective study at a single institution
Forteza et al. [12], Rev Esp Cardiol Cohort study (Level 2b)	6 patients Male: 1 (16.7%) Age: 61 [40–68] years NYHA III/IV class: 6 (100%) Two patients were being evaluated for a heart transplant. There were no patients with ventricular aneurysm and ventricular tachycardia. LVEDd, mm 60.5 [44.8–70.7] LVESV, mm 21.1 [15.4–24.9] LVEF: 68 [61–78] Maximum midventricular thickness 26 [22.5–28.2]	Median follow-up: 18 months [IQR, 6–24 months] 60.5 [44.8–70.7] 98.5 [76.1–141.0] Clinical symptoms	NYHA III/IV class: 1 (16.7) The echocardiographic study, performed after 6 months in 5 patients and after 1 month in 1 patient, showed a significant increase in ventricular volume in all patients. LVED, mm 98.5 [76.1–141.0] LVESV, mm 42.2 [28.0–66.7] LVEF, % 55 [44–66] Maximum midventricular thickness 16.5 [13.7–19.2]	Apical myectomy is a safe and effective technique for treatment and may offer an alternative to heart transplants in patients with advanced heart failure.
Schaff et al. [13], J Thorac Cardiovasc Surg Cohort study (Level 2b)	44 patients Male: 29 (66%) Age: 50 ± 17 NYHA III/IV class: 40 (91%) LVEF: 72% ± 8% LVEDV: 55 ± 17% mean indexed SV: 39 ± 17 mL/m^2^ Ventricular tachycardia n (%): 5 (11)	Operative mortality, Follow-up 1-year survival 5-year survival 10-year survival	2 (4.5%) 2.6 ± 3.1 years (median, 1.3 years) NYHA I/II class: 33 (85%) Functional status was available in 33 (85%) of 39 alive patients 95% 81% 81% LVEF: 61 ± 12% LVEDV: 68 ± 18% mean indexed SV: 46 ± 13 mL/m^2^	Transapical ventricular myectomy to increase LVEDV improves diastolic function and SV.

Abbreviations: HCM, hypertrophic cardiomyopathy; NYHA, New York Heart Association; LVEDd, left ventricular end-diastolic diameter; LVESV, left ventricular end-systolic volume; LVEF, left ventricular ejection fraction; LVEDV, left ventricular end-diastolic volume, SV, stroke volume. * Pubmed search strategy: (hypertrophic obstructive cardiomyopathy [All Fields] OR apical myectomy [All Fields]) NOT (animal[mh] NOT human[mh]) NOT (comment[All Fields] OR editorial[All Fields] OR meta-analysis[All Fields] OR practice-guideline[All Fields] OR review[All Fields] OR pediatrics[mh]).

**Table 2 life-14-00125-t002:** Apical cardiomyopathy descriptions.

Author, Date, Journal, Country, and Study Type (Level of Evidence)	Patient Group	Outcomes	Key Results	Comments
Eriksson et al. [5], J Am Coll Cardiol Retrospective study (Level 2b)	105 patients Age: 41.4 ± 14.5 NYHA I/II/III and IV class: 64/41/0 Exercise-induced ventricular fibrillation n (%): 1 (0.9)	Mean follow-up of	13.6 ± 8.3	Apical HCM in North American patients is not associated with SCD and has a benign prognosis in terms of cardiovascular mortality. One-third of these patients experienced serious cardiovascular complications, such as myocardial infarction and arrhythmias.

		Cardiovascular mortality	1.9% (2/105)	

		Annual cardiovascular mortality	0.1%	

		Atrial fibrillation	12%	

		Myocardial infarction	10%	

		Probability of survival without morbid events at 15 years	74%	
Webb et al. [4], J Am Coll Cardiol Retrospective study (Level 2b)	26 patients Age: mean 45 (range 15 to 72) Male: 20 (76.9%) Asymptomatic patients (n): 10 Atypical chest pain (n): 10 Angina (n): 6 Palpitations (n): 8 Exertional dyspnea (n): 5 Presyncope (n): 4 Fatigue (n): 4 Ventricular tachycardia n (%): 2 (7.7)	Mean follow-up 7.3 ± 6.2 years		The prognosis appeared relatively favorable in most but not all patients with apical HCM: no deaths occurred in a cohort of 26 patients over the mean follow-up period.
		Stable condition (n)	21	
		Myocardial infarction (n)	1	
		Angina (n)	2	
		Atrial fibrillation (n)	2	
Towe et al. [14], Congenit Heart Dis. Cohort study (Level 2b)	71 patients Male: 45 (63%) Age: 44.4 ± 19 years) Left ventricular wall thickness: 19.8 ± 6 mm Apical Aneurysm n (%): 4 (6)	Mean follow-up was 5.5 years (range 0.1–18.2 years)	3 patients were lost to follow-up	
		NYHA class I/II (n)	52	
		NYHA class III/IV (n)	16	
		Death (n)	14	
		Cardiac death (n)	2	
		SCD (n)	1	
		Arrhythmia (n)	27	
		LV aneurysm (n)	2	
		Stroke (n)	1	
		Apical myectomy (n)	16	
		Pacemaker (n)	8	
		ICD (n)	21	
		ICD shock	2	
Klarich et al. [6], Am J Cardiol Retrospective study (Level 2b)	193 patients Male: 120 (62%) Age: 58 ± 17 years CAD (n): 22 (11%) Apical aneurysm (n): 6 Apical dilatation with hypokinesis (n): 23 Ventricular tachycardia n (%): 21 (11) Apical Aneurysm n (%): 6 (3)	Mean follow-up period was 78 months (range, 1–350) Death from all causes (n) Heart failure Atrial fibrillation Died	187 patients [97%] 55 patients (29%; 33 women) During follow-up, more females had heart failure (*p* = 0.001), atrial fibrillation (*p* = 0.009), or died (*p* < 0.001) than males.	Survival was worse than expected (*p* = 0.001); the observed versus expected 20-year survival was 47% versus 60%. SCD, resuscitated cardiac arrest, and/or defibrillator discharge were observed in 11 patients (6%) during follow-up. Apical HCM in this population was associated with increased mortality.
Moon et al. [7], Am J Cardiol Retrospective study (Level 2b)	454 patients Male: 316 (69%) Age: 61 ± 11 years Ventricular tachycardia n (%): There were no data Apical Aneurysm n (%): There were no data	Follow-up period (43 ± 20 months)		Patients with apical HCM with poor clinical outcomes had more advanced diastolic dysfunction, reduced myocardial contraction/relaxation properties, and increased LV filling pressure at presentation.

		All-cause mortality (n)	39 (9%)	
		MACE (n)	110 (25%)	
		Hospitalization due to heart failure (n)	88 (19%)	
		Stroke (n)	26 (6%)	
		Atrial fibrillation (n)	72 (16%)	
		Syncope or presyncope (n)	5 (1%)	

SCD, sudden cardiac death; MACE, major adverse cardiovascular events; HCM, hypertrophic cardiomyopathy; NYHA, New York Heart Association; CAD, coronary artery disease; ICD, implantable cardioverter defibrillator.

## Data Availability

Data are contained within the article.

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
