# Peer review of "Transapical Approach to Septal Myectomy for Hypertrophic Cardiomyopathy"

_life, 2024, doi:10.3390/life14010125_

Round 1
Reviewer 1 Report
Comments and Suggestions for Authors
This is essentially a case report related to an elderly patient with apical HCM and associated dyspnoea who was treated with surgical myectomy, with no operative complications and subsequent symptomatic improvement. The Background section makes mention of the possibility that apical HCM carries adverse prognostic implications: there is no real support for this in the literature, and the idea is not relevant to the idea of myectomy, which does not improve prognosis in other forms of HCM.
The literature review provided is of little interest. I wuld have been more interested in the following:-
(1) HCM is associated with energetic impairment, and I suppose this is probably true of its apical form. Therefore possible contenders for medical treatment would include myosin inhibitors (eg mavacamten) and perhexiline. Perhaps this area should be discussed.
(2) It might have been worthwhile obtaining some objective indices of cardiac performance, such as BNP levels. Was this done?
(3) Elsewhere in the world, the frequency of myectomy for obstructive HCM is declining, and the procedure was never practised for the non-obstructive form. Why should apical HCM be different?
Author Response
This is essentially a case report related to an elderly patient with apical HCM and associated dyspnoea who was treated with surgical myectomy, with no operative complications and subsequent symptomatic improvement. The Background section makes mention of the possibility that apical HCM carries adverse prognostic implications: there is no real support for this in the literature, and the idea is not relevant to the idea of myectomy, which does not improve prognosis in other forms of HCM.
The literature review provided is of little interest. I wuld have been more interested in the following:-
- HCM is associated with energetic impairment, and I suppose this is probably true of its apical form. Therefore possible contenders for medical treatment would include myosin inhibitors (eg mavacamten) and perhexiline. Perhaps this area should be discussed.
Re: Thank you for your kind remarks improving our manuscript. First of all I would marked that myosin inhibitors is just added in 2023 ESC guideline and are not yet world widely accepted. Moreover there is no evidence supporting myosin inhibitors to treat apical HCM patients, meanwhile there are some published reports that underline clinical benefit after transapical myectomy for patients with apical HCM and diastolic dysfunction. Our study summarize that available literature is still not enough to perform systematic review and met analysis to obtain strong level of evidence and class I or II recommendations, however we found this surgical approach is reproducible, safe and effective.
Published data of EXPLORER, MAVERICK, VALOR trials do not provide any information regarding type of HCM. Meanwhile these trials was focused on most representative HCM patients, such as basal or midventricular hypertrophy of interventriculum septum with or without outflow tract obstruction. Therefore we suggest that myosin inhibitors is not a case for the patients with apical HCM. Meanwhile we believe that future clinical trials should address this field. Other perspective drug therapies such as perhexiline, aficamten (Cytokinetics), ninerafaxstat IMB-101 (Imbria Pharmaceuticals), Enteresto/Yuperio (Novartis), Trientine (Univar Solutions) is on the way to obtain clinical results and do not included to guideline directed medical therapy yet.
Changes: we have added this discussion in the respective section.
(2) It might have been worthwhile obtaining some objective indices of cardiac performance, such as BNP levels. Was this done?
Re: unfortunately, NT-pro-BNP levels was not routine in our practice. We do perform such indices in questionable cases, for example in patients with moderate heart valve disease, enlarged aorta and etc. to select patients for early surgery. Meanwhile
Changes: none
(3) Elsewhere in the world, the frequency of myectomy for obstructive HCM is declining, and the procedure was never practised for the non-obstructive form. Why should apical HCM be different?
Re:
In common we agree with you that septal myectomy procedure is disappearing in westernized countries, giving the way to less invasive procedures such as alcohol septal ablation. However, not for any case this statement is true. In high volume HCM centers, such as Mayo clinic septal myectomy is a gold standard procedure, meanwhile less invasive alcohol septal procedure is only an alternative option for high risk patients, that is consistent with current ESC and ACC/AHA guidelines.
In our case current medications, such as non-vasodilating beta-blockers, was ineffective, and after shared decision-making, including the risks and benefits of all treatment options, transapical septal myectomy was considered an alternative to escalation of medical therapy.
We are confident that alcohol septal ablation is not a case for apical HCM. This procedure altered first septal branches that may supply basal and some parts of middle part of interventricular septum, however do not affect apical part. Moreover, this procedure is focused on resolving obstruction of LVOT that was not a case in our patient.
Changes: respective changes have been made in the Discussion.
Reviewer 2 Report
Comments and Suggestions for Authors
In the study by Alexander Afanasyev et al., they investigated “Transapical Approach to Septal Myectomy for Hypertrophic Cardiomyopathy”.
This was an very interesting case report on whether cardiac myectomy was effective for symptomatic heart failure caused by diastolic disfunction on apical hypertrophic cardiomyopathy. However, there were several concerns that need to be addressed.
Concern #1
It was necessary to ensure with high quality whether this patient’s simptons such as shortness of breath was an diiastolic disfunction due to cardiac hypertrophy. Your data of echocardiography did not describe such as deceleration time, E/E’ and left atrial strain as indicators of diastolic disfunction.
Please include these parameters as diastolic function of echocardiography. In addition, please discribe the results of exercise stress echocardiography and cardiac catheterization, if any.
Concern #2
This question is related to Concern #1. NYHA classification and value of 6-minute walk were shown after surgery, however did you not evaluate before surgery? You should also describe the preoperative and postoperative changes in the indicators of diastiolic disfunction by echocardiacgraphy. It is important to demonstrate improvement in heart failure using objective indicators.
Concern #3
Was ethical consideration made when choosing this treatment that has not been described in this guideline? Please clearly state the ethical considerations.
Concern #4
In Table 1, the details of information you collected were not enough. Generally, cardiac myectomy for apical hypertrophic cardiomyopathy has been performed for ventricular aneurysm and uncontrollable ventricular tachycardia. Please summarize the detail of surgical indications for apical hypertrophic cardiomyopathy that you have collected in every papers.
Concern #5
Did the pathological findings of an enough amount of myocardium match hypertrophic cardiomyopathy? It is important to exclude amyloidosis and Fabry disease in the differential diagnosis of cardiac hypertrophy. You should show whether there were pathological findings specific to hypertrophic cardiomyopathy.
Author Response
In the study by Alexander Afanasyev et al., they investigated “Transapical Approach to Septal Myectomy for Hypertrophic Cardiomyopathy”.
This was an very interesting case report on whether cardiac myectomy was effective for symptomatic heart failure caused by diastolic disfunction on apical hypertrophic cardiomyopathy. However, there were several concerns that need to be addressed.
Concern #1
It was necessary to ensure with high quality whether this patient’s simptons such as shortness of breath was an diiastolic disfunction due to cardiac hypertrophy. Your data of echocardiography did not describe such as deceleration time, E/E’ and left atrial strain as indicators of diastolic disfunction.
Please include these parameters as diastolic function of echocardiography. In addition, please discribe the results of exercise stress echocardiography and cardiac catheterization, if any.
Re: Thank you for your kind remark. There were no comorbidities causing shortness of breath. Moreover, in retrospective view we found improvement on functional capacity after transapical myectomy, that may explain association between patient’s symptoms and diastolic dysfunction due to apical hypertrophy.
Unfortunately, novel indices of left ventricle diastolic function including left ventricle global longitudinal diastolic strain rate and left atrial systolic strain measurements do not perform by routine in our Centre, therefore they have not been assessed in this case. We classified diastolic dysfunction as grade II by increased left atrial volume index (44ml/m2), average E/e’ > 14, reduced stroke volume index (22ml/m2) and preserved left ventricle ejection fraction (69%).
We routinely perform echocardiography with upright exercise. As we stated, maximum provokable LVOT gradient was 15 mmHg. We did not found indications to perform supine bicycle or pharmacological stress testing. In routine practice in HCM patients with basal or midventricular left ventricle outflow tract obstruction we do perform Dobutamine infusion and/or atrial pacing intraoperatively to assess result of transaortic septal myectomy with invasive measurements. However this was not a case.
We do not perform cardiac catheterization to diagnosis apical HCM. Echocardiography and CMR imaging were enough.
Changes: we have added echo parameters described above.
Concern #2
This question is related to Concern #1. NYHA classification and value of 6-minute walk were shown after surgery, however did you not evaluate before surgery? You should also describe the preoperative and postoperative changes in the indicators of diastiolic disfunction by echocardiacgraphy. It is important to demonstrate improvement in heart failure using objective indicators.
Re: thank you for your remark improving our manuscript. The patient was in NYHA class III, however preoperatively shortness of breath requiring interrupt 6min walk test occurred while walking a 50m distance, that may be considered to more severe functional impairment, near to NYHA class IV.
Changes: respective changes have been done in main text
Concern #3
Was ethical consideration made when choosing this treatment that has not been described in this guideline? Please clearly state the ethical considerations.
Re: this is very good question, thank you! There were two opposite views: escalation of medical therapy that was currently ineffective, other one is surgery. After a literature review and after shared decision-making, including the risks and benefits of all treatment options, transapical septal myectomy was considered.
Changes: Ethical statement has been clarified.
Concern #4
In Table 1, the details of information you collected were not enough. Generally, cardiac myectomy for apical hypertrophic cardiomyopathy has been performed for ventricular aneurysm and uncontrollable ventricular tachycardia. Please summarize the detail of surgical indications for apical hypertrophic cardiomyopathy that you have collected in every papers.
Re: thank you for your valuable comment, we have added surgical indications in the table
Changes: respective changes have been done.
Concern #5
Did the pathological findings of an enough amount of myocardium match hypertrophic cardiomyopathy? It is important to exclude amyloidosis and Fabry disease in the differential diagnosis of cardiac hypertrophy. You should show whether there were pathological findings specific to hypertrophic cardiomyopathy.
Re: There were no clinical or CMR with contrast imaging data suggesting amyloidosis, Fabry disease or other lysosomal storage disease in this case. Therefore we did not perform DNA screening for these conditions.
Changes: we have added statement, that storage diseases were excluded.
Round 2
Reviewer 2 Report
Comments and Suggestions for Authors
Alexander Afanasyev et al addressed most of the comments in their current version of revision.
However, important issue for publishing this article have not been resolved.
In the literatures you collected, it seems that there has been very few cases of surgery due to heart failure symptoms (many were unknown reasons)
What was important here was to prove heart failure simpton of diastolic dysfunction very carefully from strict objective data. The e/e’ at rest was borderline. It was not possible to judge the patient’s shortness of breath as a symptom of HFpEF, and the evidence for choosing a invasive treatment was extremely important.
You explained that there has been no evidence of effect for drug treatment, however if the evidence level was equivalent, it was recommended to choose a less invasive treatment.
Thus, what was most important was to clarify the process including ethical considerations for choosing this treatment (your description of ethical considerations was insufficient).
Did you have the consent of a heart failure specialist for your diagnosis? Have you explained to the patient that the treatment effect is unknown? For example, myocardial size reduction due to myocardial or ventricular aneurysm resection and constrictive pericarditis as a complication may worsen diastolic dysfunction. Did you get the approval of the hospital ethics committee for your selection of operation?
Author Response
Comments and Suggestions for Authors
Alexander Afanasyev et al addressed most of the comments in their current version of revision.
However, important issue for publishing this article have not been resolved.
In the literatures you collected, it seems that there has been very few cases of surgery due to heart failure symptoms (many were unknown reasons)
What was important here was to prove heart failure simpton of diastolic dysfunction very carefully from strict objective data. The e/e’ at rest was borderline. It was not possible to judge the patient’s shortness of breath as a symptom of HFpEF, and the evidence for choosing a invasive treatment was extremely important.
You explained that there has been no evidence of effect for drug treatment, however if the evidence level was equivalent, it was recommended to choose a less invasive treatment.
Thus, what was most important was to clarify the process including ethical considerations for choosing this treatment (your description of ethical considerations was insufficient).
Did you have the consent of a heart failure specialist for your diagnosis? Have you explained to the patient that the treatment effect is unknown? For example, myocardial size reduction due to myocardial or ventricular aneurysm resection and constrictive pericarditis as a complication may worsen diastolic dysfunction. Did you get the approval of the hospital ethics committee for your selection of operation?
Re: Dear Reviewer, thank you for your kind remarks that underline your high level of the expertise. We agree with you that available literature data regarding apical HCM prognosis, natural history, pharmacological and surgical management are very limited.
We had no doubt that in this case the symptoms of heart failure were associated with left ventricle diastolic dysfunction, that was documented in a medical history. In fact, in this patient retrospectively we did not find exact e/e’ ratio in the echocardiography routine protocol. Unfortunately in medical report we found only the statement that patient had grade II LV diastolic dysfunction, therefore we have indicated only e/e’ borderline. That is an unnecessary and annoying limitation of our case report.
We have obtained informed consent for the surgery with detailed discussion regarding risk of surgery, including worsening diastolic dysfunction, heart failure symptoms, death, etc.. Transapical myectomy have been included in the standard of care of HCM patients by the Ministry of Health of the Russian Federation, therefore the approval of the local ethics committee for the selection of operation was not required.
We do not agree that it would be better to choose a less invasive treatment in this case. Patient was highly symptomatic NYHA class III to IV. Guideline directed medications was not effective. Alcohol septal ablation is not a case for apical HCM. The effect on novel myosin inhibitors in apical HCM subset is unknown and they are still not available/approved in Russia. Despite the fact that early reports support benign prognosis, in contrast relatively high volume study [6] has underlined up to 29% death rate at mean follow-up 6 years, moreover with worse result in women then in men (45% vs 19%, p<0,001). Furthermore, NYHA class III-IV, left atrial enlargement have been indicated as independent predictors of cardiovascular morbidity, along with high e/e’ ratio. After shared decision-making, including the risks and benefits of all treatment options, transapical septal myectomy was considered an alternative to escalation of medical therapy in this clinical scenario.
We used to recommend conservative management and dynamic evaluation due to expected benign prognosis, but since 2023 we have considered the surgical treatment for severely symptomatic apical HCM patient.
Our Institute have become the highest volume HCM-center in Russia, currently we perform more than hundred septal myectomies annually with more than 1000 procedures in past ten years. In our heart team also obligatory included heart failure specialist, heart transplant specialist, experienced HCM-cardiologist and surgeons, ECHO and CMR specialists. They agreed that the apical HCM is a true diagnosis in this patient, they also have excluded other reasons that may explain NYHA class III-IV heart failure symptoms. Due to absence of own experience of transapical myectomy for apical HCM patients, we have discussed the diagnosis and management with Hartzell Schaff and colleagues from Mayo clinic and got support for transapical myectomy as recommended way to manage the apical HCM in this patient (as well as following apical HCM patients in 2023). We have underlined the role of H. Schaff in the Acknowledgments section. Meanwhile we should mention, that transapical LV approach was not something new for our surgical practice, we have own experience of more than 800 LV postinfarction apical aneurysm repairs (Dor, Cooley, Jatene, Stoney, etc.).
Dear reviewer, thank you for your valuable comments and useful suggestions, that definitely improved our manuscript.